



# Ideas and perspectives: Beyond model evaluation - combining experiments and models to advance terrestrial ecosystem science

Silvia Caldararu[1,2], Victor Rolo[3], Benjamin D. Stocker[4,5], Teresa E. Gimeno[6], Richard Nair[1,2]

[1] Discipline of Botany, School of Natural Sciences, Trinity College Dublin, Ireland

[2] Max Planck Institute for Biogeochemistry, Jena, Germany

[3] University of Extremadura, Forest Research Group, INDEHESA, Plasencia, Cáceres, Spain

[4] Institute of Geography, University of Bern, Bern, Switzerland

[5] Oeschger Centre for Climate Change Research, University of Bern, Bern, Switzerland

[6] CREAF, 08193 Bellaterra (Cerdanyola delVallès), Catalonia, Spain

*Correspondence to*: Silvia Caldararu (caldaras@tcd.ie) and Victor Rolo (rolo@unex.es)

**Abstract.** Ecosystem manipulative experiments are one of the most powerful tools to understand terrestrial ecosystem responses to global change because they measure real responses in real ecosystems. However, their scope is limited in space and time due to cost and labour intensity. This makes generalising results from such experiments difficult, which creates a

conceptual gap between local scale process understanding and global scale future predictions. Recent efforts have seen results from such experiments used in combination with dynamic global vegetation models, most commonly to evaluate model predictions under global change drivers. However, there is much more potential in combining models and experiments. Here, we discuss the value and potential of a workflow for using ecosystem experiments together with process-based models to enhance the potential of both. We suggest that models can be used prior to the start of an experiment to generate hypotheses,

identify data needs and in general guide experimental design. Models, when adequately constrained with observations, can also predict variables which are difficult to measure frequently or at all, and together with the data can provide a more complete picture of ecosystem states. Finally, models can be used to help generalise the experimental results in space and time, by providing a framework in which process understanding derived from site-level experiments can be incorporated. We also discuss the potential for using manipulative experiments together with models in formalised model-data integration

frameworks for parameter estimation and model selection, a path made possible by the increasing number of ecosystem experiments and diverse observation streams. The ideas presented here can provide a roadmap to future experiment - model studies.



## 1 Introduction


In the face of a changing climate, understanding how ecosystems will respond to conditions unprecedented in the observational record is one of the biggest challenges in earth system science. To meet this challenge, we must employ all tools available to us - observations, experiments and meta-analyses, and process- and data-based models. All these different tools have strengths and weaknesses in terms of spatial and temporal scales, generality and predictive capacity. Therefore, not only using these

tools, but using them in combination, is essential for bridging the gap between process understanding and scalability.

Over the last few decades, there have been great advancements in the type and quantity of available ecosystem data, including large databases of field observations (Kattge et al., 2020); continuous ecosystem monitoring (Baldocchi, 2020) remotely sensed Earth observations (Xiao et al., 2019); spatial datasets with global coverage for climate (Fick & Hijmans, 2017; Harris et al.,

2020; Karger et al., 2020), soil (Food and Agriculture Organization & Food and Agriculture Organization of the United Nations, 2008; Hengl et al., 2017),  and other spatially upscaled global ecosystem data products (Jung et al., 2020; Poggio et al., 2021). In and of themselves, such observational datasets provide a comprehensive picture of ecosystem properties and structure, carbon and water fluxes. However, the mounting data richness in ecosystem science has not led to a reduction in projection uncertainties of global biogeochemical cycles in a future climate (Arora et al., 2013; Friedlingstein et al., 2014).

Ecological and physiological process understanding - key for predicting outside the domain of observed conditions - often remains weakly constrained in spite of the large volume of  available ecosystem data collected over recent decades. In this context, ecosystem manipulation experiments (EMEs) have a unique and particularly valuable role to play (Stocker et al., 2016; Wieder et al., 2019).

EMEs are outdoor experimental setups in which one or multiple driving factors (e.g., water availability, temperature, nutrient inputs, atmospheric $CO_2$ concentration, perturbations, species composition), or a combination of factors, are controlled to study their separate or interactive effects on ecosystem processes as well as subunits within ecosystems such as plants or microbes. They are a useful tool because they contain ecosystem process responses defined by treatments and hypotheses, while minimising confounding factors, and scaling biases and experimental artefacts from smaller and more limited lab or

mesocosm-based experiments (Dalling et al., 2013). EMEs provide a unique window into the responses of ecosystems to a potential future environment and can reveal complex responses at the level of plants, communities, and ecosystems, shaped by feedbacks between soil and vegetation. While extremely valuable for understanding ecosystem responses to future conditions, due to their realistic scale and multidimensionality of observations, EMEs generate small data - in contrast to, e.g., Earth observation. Therefore, it remains challenging to generalise from a relatively small set of local observations to future global

predictions. EMEs are usually operated at spatial scales on the order of 1-100 m, and are limited in time to typically 1-10 years due to high operating costs and laborious measurements. Thus, it can be difficult to scale the conclusions from experiments with confidence, both in time and in space. Here, we propose and discuss approaches for aiding generalisations from EME observations in combination with process-based models as a potential solution to overcome this scalability challenge.



Process-based ecosystem models are mathematical representations of how plant traits, and soil characteristics determine water, energy, and biogeochemical fluxes, and the accumulation of organic matter in a cascade of ecosystem pools, given a set of environmental boundary conditions. Depending on the scale and processes represented, such models are termed terrestrial biosphere models, dynamic vegetation models, mechanistic ecosystem models, or land surface models, and are implemented as coupled components in Earth System Models (Prentice et al., 2007). Irrespective of what we call them, all such models face

similar challenges, in that they need to balance realism, robustness, and reliability (Prentice et al., 2015)  to achieve sound predictions for ecosystem fluxes and pools across the globe and across decades into the future. To achieve this such models need to be confronted with diverse data to leverage complementary constraints. Recent efforts have been made in using large observational datasets for validating and benchmarking these models (Collier et al., 2018; Seiler et al., 2022). However, as outlined above, model projections remain divergent ((Arora et al., 2013; Friedlingstein et al., 2014)), pointing to remaining

gaps in critical process understanding from such datasets and extrapolating observations from past climate conditions to the future often comes with increased uncertainty.

EMEs have been used to parameterise, validate and test models, ranging from drought (McDowell et al., 2013; Powell et al., 2013), warming (Ma et al., 2022; Parton et al., 2007; Zaehle et al., 2010), fertilisation (Meyerholt & Zaehle, 2015; Thomas et

al., 2013), or decomposition (Bonan et al., 2013) experiments. A step change came about with the FACE Model Data Synthesis (MDS) project (De Kauwe et al., 2014; Medlyn et al., 2015; Walker et al., 2014; Zaehle et al., 2014) which synthesised data from two temperate forest Free Air Carbon Dioxide Enrichment EMEs. The project asked not only if the models match the data but why the models match the data, making full use of manipulative experiments and being able to plot a path to model improvement. This approach led to the identification of both key processes that need to be included in models to correctly

capture responses to elevated $CO_2$ (e.g., flexible but realistic tissue stoichiometry, flexible biomass allocation, flexible leaf mass per area, organic N uptake) alongside areas where more data is needed to identify realistic and reliable process representation (e.g. water limitation effects on photosynthesis and transpiration, wood turnover, detailed photosynthesis information, N losses). Other studies are now following up on these results to further improve models (Caldararu et al., 2020). Since then, there have been several other MDS studies following the same philosophy, including other FACE experiments

(Fleischer et al., 2019; Medlyn et al., 2016) and FACE plus warming experiments (De Kauwe et al., 2017). A few such modelling studies were performed prior to the start of the actual experiment (Fleischer et al., 2019; Medlyn et al., 2016; Parton et al., 2007) with the aim of informing experimental design and in the case of (Parton et al., 2007). Their model predictions were later tested against experimental results (Dijkstra et al., 2010), showcasing the power of such a study.

There has been an increase in the number of EMEs globally, as well as in the amount of open data and efforts to synthesise and bring together existing experiments (Cleland et al., 2019; Liang et al., 2020; Van Sundert et al., 2023). The present issue includes some examples of new experiments including a large-scale drought and fertilisation experiment (Vargas G. et al.,



2022) and EME-model integration, including studies looking at new nutrient models (Cornut, Delpierre, et al., 2022; Cornut, le Maire, et al., 2022; Eastman et al., 2023; Li et al., 2022), the effects of root distribution of water availability response (Kulmatiski et al., 2023), carbon storage in grasslands (Wilcox et al., 2022) and extreme events (Holm et al., 2022).

Global observations, ecosystem experiments, and process-based models all have their strengths and weaknesses in terms of scalability and process understanding (Figure 1). To advance ecosystem science an integration of complementary sources of observations and models is needed to achieve the full predictive potential of the tools available. The majority of EME-model integration studies have focused on constraining or evaluating models with experimental data, however, there is a large untapped potential to use models and EMEs bidirectionally, as they can bridge the process understanding - scalability gap. While models are capable of making global scale predictions on long timescales, they also include a detailed process representation which can be a reflection of the insight gained from EMEs. In this paper we set out a roadmap to using models together with EMEs not only as a one-way street for validating models but as an integrated workflow for improving experimental design and generalising experimental conclusions.

## 2 Not a one sided relationship: what can models do for experiments?

In past studies, the focus has been on integrating conclusions from EMEs into models and evaluating such models against experimental data to improve model process representation (Medlyn et al., 2015; Norby et al., 2016). However, this relationship can easily be turned around. Models can be used as digital twins or sandboxes to explore different possibilities for experimental design, set realistic and practical treatment levels and formulate quantitative, defensible hypotheses. Models can also be used to integrate observations gained from one or more experiments for generalising in time and space

### 2.1 Hypothesis generation and experimental design

Ecosystem manipulation experiments are designed around testable research questions and hypotheses, following established theory. Hypotheses are generally of a qualitative and first-order nature, i.e., 'if higher A then B', for example nutrient addition leads to increased growth and photosynthesis or drought reduces growth. However, ecosystem responses are affected by multiple feedbacks, operating at diverse time scales. Responses can be non-linear, even threshold-like, and interactive, and can be affected by plant adaptations, species-specific responses, and plant-soil interactions. Certain processes saturate at higher levels reached over time, such as nutrient addition or elevated $CO_2$, where continued application or the treatment does not lead to a further ecosystem response, as systems become limited by other factors. In the case of edaphic factors, such as nutrient availability, spatial factors, such as underlying fertility and history of deposition interact with future loading. Ecosystem responses have also been shown to vary with the intensity and duration of the treatment (Niu et al., 2022). This means that it is often difficult to (explicitly) generate hypotheses that reflect multiple known individual processes and their (unknown) interactive effects. Using a model, or multiple models, or multiple process representations within one modelling framework can aid the hypothesis generation.



As many responses to environmental drivers are non-linear, it can be uncertain when designing an experiment what and how many treatment levels are necessary, especially if choosing a response surface rather than a replicated approach (Kreyling et al., 2018) This is particularly important as ecosystem-level experiments are technically challenging and expensive, especially those on the scale of FACE experiments. There are financial, logistical, and practical constraints on how many treatment levels can be considered. Models can easily be run with multiple treatment levels, which would allow identification of key nonlinearities and a defined response surface which can be used as a basis for choosing experimental treatments.

Models can also potentially be used to define a sampling strategy. Initial model simulations can help identify which variables are likely to respond to the experimental treatment and at what time scales treatment responses are likely to be detected. If using multiple models or a model with multiple alternative process representations, this can also help identify variables that would help test the proposed hypotheses, by looking at model variables that respond differently for different model formulations. We propose a workflow (Figure 2) in which one or multiple models are run with local site conditions with multiple treatment levels. This then allows us to identify hypotheses based on predicted responses, ideal treatment levels and key variables to measure. This can be done multiple times to explore different experimental setups and ideas.

There is often a gap between the optimum sampling strategy and what is actually feasible in field experiments - variables often not be measured with a sufficient temporal or spatial frequency due to either costs or technical difficulties. While we as a community have come a very long way in automating certain measurements - carbon and water fluxes, spectral properties, soil moisture - many remain time and labour intensive, such as direct measurements of biomass above and below- ground or plant and soil chemical composition. Even more, other quantities which might be useful to know are not possible to measure, that is they are latent or conceptual variables such as plant nutrient demand. Such variables are often key in explaining the observed responses, but can usually only be guessed at.

In contrast, models produce all variables consistently at daily or sub daily intervals and predictions, including conceptual or latent quantities which can be expressed mathematically can therefore be used alongside experimental results to explain observed responses. For example, the issue of nutrient limitation under elevated $CO_2$ involves many plant and soil processes, many of which cannot be monitored easily but are included in nutrient-enabled models (Caldararu et al., 2022). One potential issue here is that models need to be able to accurately represent ecosystem processes at a given particular experimental site. Before being used in such a scenario, models should be evaluated or even calibrated for the site to make sure of the model performance.

## 2.2 Generalising in time and space

Most manipulative ecosystem experiments are commonly performed at the plot level and are only maintained for a short time period (1- 5 years), with a few exceptional experiments lasting in the range of ten years or more (Eastman et al., 2021; Magill et al., 2004). While a particular experiment can tell us a lot about the mechanistic process involved, it remains an open question how scalable they are in time and space. While there are increasing efforts to standardise experimental treatments through initiatives such as the Nutrient Network (Borer et al., 2014), only a small and inevitably imperfectly distributed area of the





Earth's vegetated surface can ever be covered by experimental treatments. However, models can be run over relatively long periods of time and globally, helping to test the generality of the process understanding informed by EMEs. .

The problem of spatial scaling is obvious. Can the particular ecosystem studied represent other ecosystems of the same type and can plant or soil processes inferred from observed response be generalised? Such upscaling has been previously performed by accounting for moderating factors of the response, such as climate or soil conditions (Terrer et al., 2021). However, such approaches lack the mechanistic and dynamic underpinning provided by a process-based model and could thus miss important feedbacks.

A particular challenge is posed by the step shape of imposed environmental change in EMEs, which contrasts with the gradual decadal-scale environmental change to which ecosystems are exposed. Several processes make the responses to a continuous change potentially different from those to a step change. Plants acclimate to the new conditions over time (Reich et al., 2018), a process that has been primarily studied for changes in temperature but is likely to occur for most environmental drivers (see Section 5). Species composition is likely to change under different environmental conditions, but it can likely only be observed during the duration of the experiment in systems with high species turnover such as grasslands (Avolio et al., 2014). Although ecosystem responses in observable quantities can often not be assumed to be time-invariant, particularly if slow plant-soil feedbacks are involved and even if conditions were held constant after a step-change, underlying processes often can. Mechanistic models that simulate responses to step changes, and that are constrained by data from step-change experiments are thus a device for translating empirical insights gained from step-change experiments into predictions that respect *a priori* knowledge of slow processes and their role in shaping decadal-scale ecosystem dynamics.

A challenge for model-enabled temporal upscaling are slow processes that are not represented in models or insufficiently constrained by relatively short-term observations (1-10 years in EMEs), but can modify ecosystem responses at the longer term. Acclimation of photosynthesis and respiration to changes temperature and $CO_2$ (Prentice et al., 2014; Way & Yamori, 2014) and plasticity in allocation in response to carbon-nutrient balance changes (Poorter et al., 2012; Zaehle et al., 2014) largely lack reliable representations in models, and may have a strong influence on decadal to centennial carbon, nitrogen, and phosphorous supply and demand, and ecosystem dynamics. On even longer timescales, genetic adaptation through selection under new environmental pressures and changes in species compositions may emerge as strong drivers of ecosystem responses. Some experiments on shorter-lived, herbaceous species have shown shifts in species composition in response to elevated $CO_2$ (Reich et al., 2018), similar changes in longer-lived organisms are harder to observe in EMEs. These types of responses and their lack of representation in ecosystem models imply limits for data-constrained and model-based temporal up-scaling. In terms of short-term plasticity or acclimation, recent advances in using eco-evolutionary optimality in models (Harrison et al., 2021) can represent plastic plant responses, and have successfully been used together with EMEs (Caldararu et al., 2020; Sabot et al., 2022). However, current ecosystem models are not well suited for dealing with changes in species composition, demography, and competition processes. However, trait-based, individual or cohort-based models (Fisher et al., 2015). Particularly for evaluating simulated acclimation and other phenotypic plasticity that operate at time scales of months to years, it will be important to make targeted use of insights gained from EMEs. Since this is a knowledge gap in both EMEs and





models, it is an opportunity for both communities to work side by side, rather than sequentially, to advance our knowledge of plant adaptation.

In recent years, there has been a global effort to overcome the limitations of the different approaches, and try to solve the need
of standardised controlled experiments on wide temporal and spatial scales. (Fraser et al., 2013) coined the term "coordinated distributed experiments" (CDEs) to describe a global network of standardised experiments distributed across a wide range of ecosystems and climate zones to account for issues of spatial and temporal scales. While initial CDEs were limited in their spatial coverage to Europe or North America, recent ones are striving to be truly global. Examples of global CDEs include the International Drought Experiment (Knapp et al., 2017) that studies the sensitivity of ecosystems to extreme drought events,
the Nutrient Network (Borer et al., 2014) which addresses how grasslands are affected by eutrophication and grazing or the Tree Diversity Network (Paquette et al., 2018) which includes a global collection of tree biodiversity experiments. Coordinated EME networks with standardised protocols and homogenised compilations of published EME data (Van Sundert et al., 2023) can drastically facilitate model-data integration studies, performed for an extended set of sites and experiments. Multi-experiment modelling may be essential for powerful generalisability tests and uncertainty quantification of model predictions,
e.g., by a leave-experiment-out cross-validation (see also Section 3). Thus, using experiment networks and compilations together with models would increase confidence in the generality of model predictions and partially deal with the upscaling issues discussed in Section 3.3. The use of CDE results could also help to improve the parametrization of process-based models in ecosystems that are underrepresented, but this would require better coordination between response variables measured in the field and those processes included in the models (N. G. Smith et al., 2014).

## 3 Model-data integration

Data assimilation (DA) and model data integration (MDI) are broad umbrella terms for a variety of statistical methods that fit process based models parameters to observations. The methods are well established and widely used with remote sensing (Exbrayat et al., 2019; W. K. Smith et al., 2020) and eddy covariance data (Fox et al., 2009; Kuppel et al., 2012), at both site and global scales. There are now a number of DA methods available to the community (Anderson et al., 2009; Fer et al., 2021;
Huang et al., 2019), which significantly lowers the barrier to entry for those who want to use such tools.
One of the main issues with using DA methods is that the data used to constrain models are observations in present or past conditions, raising questions about the capacity of resulting models to predict ecosystem responses under future conditions. Therefore, using data from manipulative experiments can be extremely valuable in providing information of yet unobserved conditions. One other common issue with using remote sensing data to parameterise models, is that the observations used in
DA need to be variables that are actually represented in models (MacBean et al., 2022), so that most remote sensing indices need to be processed further before they can be used. In contrast, experimental observations provide information that can easily be mapped to model variables - biomass, ecosystem fluxes, soil pools, etc.



MDI provides a formalised approach to make best use of naturally sparse EME observations and combine them with *a priori* understanding embodied in model structures. Integrating diverse ecosystem data can help estimating the system state, given

physical constraints that are built into the model (e.g., mass conservation) (Jiang et al., 2020). MDI can also provide an approach to formalised model selection (Mark et al., 2018) and a systematic treatment for trading off model complexity and fit to the data.. EMEs, in contrast with observational data commonly used in MDI studies, often have a particularly strong leverage in discriminating between predictions of alternative models, which other data types often lack. Only if an ecosystem's slow biogeochemical cycling is "hit hard" in an experimental setup, underlying processes are revealed. Thus, EMEs provide

key information that is required for robust model selection - the discrimination of alternatively formulated model structures that reflect alternative hypotheses of how ecosystem processes work.

However, typical models used for global biogeochemical cycle and climate change impact simulations are often complex and contain a large number of weakly constrained parameters. In view of the sparsity of EME data, this poses a risk of overfitting.

Over-fitting may be mitigated using a "leave-experiment-out cross-validation" approach where one experiment is systematically left out of the model fitting procedure and used as an out-of-sample test. This may be a way to handle the overfitting risk, enable a more robust calibration of model parameters, and provide a more reliable estimate of the spatial generalisation error. However, the environmental space currently covered by EMEs and their available data is limited and gaps remain particularly in the tropics and for $CO_2$ experiments in all biomes except temperate forests and grasslands (Van Sundert

et al., 2023).

## 4 New data for EME-model synthesis

Despite the potential for model-informed ecosystem experiments discussed above, data availability still defines model use because without data, models are impossible to constrain. Advances in measurement techniques and data processing offer an opportunity to increase the types and frequency of measurement that can be gathered at EMEs. EMEs are one of the best

locations to develop new data streams because they are typically already well parameterised, offering established data in addition to that from novel sources and potentially benefiting from frequent site visits necessary for non-standard instrument development. Thus in addition to the direct feedback with individual EMEs, models can set the agenda for data development through transparent discussion of the uncertainty and parameter sensitivity.

Perhaps the most straightforward, EME experiments can be equipped with proximal sensing devices such as phenocams

(Brown et al., 2016) or sun induced fluorescence (SIF) sensors (Yang et al., 2015). These provide continuous regular measurements of (generally) canopy properties. While these types of measurements are available from spaceborne sensors, local measurements provide more spatial detail and often the possibility of measuring each individual plot separately, thus identifying treatment effects in smaller scale EMEs. Unlike spaceborne instruments, proximal sensors usually require calibrating sensors between different treatments  but on the other hand provide regular, location specific information. Such





continuous measurements are extremely useful for model evaluation and development as they are often done at a temporal frequency similar to that of a model timestep and can provide information on short-term changes in vegetation responses. Additionally, such data can also be used in DA contexts (Section 3) much more easily than irregular, spot measurements. One application of note on the horizon is capturing patterns of functional diversity, which can be validated in the field (Pacheco-Labrador et al., 2022) and through EME explore causative mechanisms behind changes in environment, changes in function

and changes in species or functional diversity. As models move more towards representing species diversity (Kauwe et al., 2015) and new traits based demographic models become more widespread (Xu & Trugman, 2021), such measurements in the context of EMEs can provide an invaluable source of process understanding.

Given widespread coverage, data captured from proximal sensing devices or field measurements may also potentially be upscaled via wide networks (which do not necessarily rely on EMEs). Many modern approaches to this problem rely on

machine learning (ML) methods to reach from regional or global scales (Lapeyre et al., 2020; Poggio et al., 2021)). But in EMEs and similar contexts, ML also offers another complementary tool: data streams which are either difficult to capture or process (Nair et al., 2023) can be gap filled or interpreted with higher confidence and representativeness. Further development of techniques currently only possible at laboratory or homogenous agricultural scale may allow dynamic sub annual time series of difficult to measure parameters such as photosynthetic capacity (Heckmann et al., 2017), nutrient pools both in biomass and

available in soil (Tan et al., 2022), or phenological dynamics beyond leaves, especially those belowground (Wang et al., 2022). This is particularly relevant belowground, where data are particularly sparse. However technical and logistical limitations remain and on-site instrumentation is considerably more challenging than laboratory studies. A key challenge here is capturing both spatial and temporal dynamics at the same time. In many cases many sensor nodes are preferable because ecosystem properties in general and belowground properties in particular are very heterogeneous. However, when such technologies are

fully deployable in the field and an EME context, they can provide datastreams for models which were not previously available, reducing uncertainty in difficult to measure processes.

Eddy covariance (EC) data are one of the most frequently used datastreams for model evaluation, parameterization and data assimilation. The advantage of using EC data is its high temporal frequency and the large number of globally distributed sites

where such data is available. However, as detailed above (Fig. 1), they are purely observational measurements and lack the specific advantages of an EME. A small number of EC sites do use a treatment, albeit on an unreplicated treatment scale (El-Madany et al., 2021; Zhao et al., 2022). EC experiments are laborious to construct and expensive to operate in tall stature ecosystems, and even in shorter ecosystems the footprint of an EC tower is a challenge for application of many global change relevant treatments. Indeed, only a small number of manipulations can be conducted effectively on this scale and thus far are

limited to nutrient treatments. These designs also necessitate a well-supported baseline of major ecosystem parameters before treatments to partially substitute for replication. This is a necessary compromise versus the scale and unprecedented realism which can be achieved.

A key aspect of several modern designs is dispensing of replication. While this is controversial in the classic model of the scientific methods, as we highlighted with the EC measurements example this offers advantages in this case for realism which could not reasonably be captured in other fashions. Similarly, quantitative gradients allow parameterisation of response surfaces (as opposed to step changes in standard EMEs) but limit possible replication (Kreyling et al., 2018; Piepho et al., 2017). In a model-EME context, this allows the models to be evaluated at multiple treatment levels, circumventing the step change issues. Large, full ecosystem level experiments with sometimes groundbreaking treatments such as the TEMPEST flooding experiment (Hopple et al., 2023), are also not replicated but provide invaluable insights into ecosystem functioning. An alternative example is the unique Biosphere2 setup, which allowed a fully traceable experiment on a 'whole ecosystem', albeit without replication (Werner et al., 2021). This would allow intensive tracing of treatment effects in models through a process-by-process validation, rather than the usual validation against end results such as biomass responses.

## 5 Conclusions and outlook

Ecosystem manipulative experiments and process-based models are both powerful tools of understanding how terrestrial ecosystems are going to respond to future global change. However, on their own, they each suffer from their own particular short-comings, including lack of generality in time and space in the case of experiments and lack of process realism in the case of models. We propose that used in combination, these two tools can achieve their full potential. Going beyond a one-sided process where data from experiments is used to inform models, we lay out a roadmap (Figure 2) for using experiments and models alongside each other, and longside new types of measurements and techniques, as complementary sources of information. This approach will further our understanding of how terrestrial ecosystems work and increase our predictive capability of the future of terrestrial ecosystems.

## Acknowledgments

We would like to acknowledge the help of Dr. Sönke Zaehle and Dr. Karin Rebel for their comments on early drafts of this opinion piece. B.D.S. was funded by the Swiss National Science Foundation grant PCEFP2_181115. This work is a contribution to the LEMONTREE (Land Ecosystem Models based On New Theory, obseRvations and ExperimEnts) project, funded through the generosity of Eric and Wendy Schmidt by recommendation of the Schmidt Futures program (B.D.S.).

## Conflict of interest

The authors declare no conflict of interest.



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

| | EMEs | Observations | Models |
|---|---|---|---|
| Real Process Representation | ● | ● | ● |
| Real Process Understanding | ● | · | ● |
| Spatial Generality | · | ● | ● |
| Long Term Temporal Generality | · | ● | ● |
| Short term Temporal Specificity | ● | ● | · |
| Representative Novel Conditions | ● | · | ● |
| Reasonable Intercomparability | · | ● | ● |

Figure 1 - Strengths and weaknesses of ecosystem manipulation experiments, observations and process based models. Green areas with large circles represent areas that a particular tool is strong at, orange with medium circles areas of medium strength and red with small circles areas where each tool is weak.



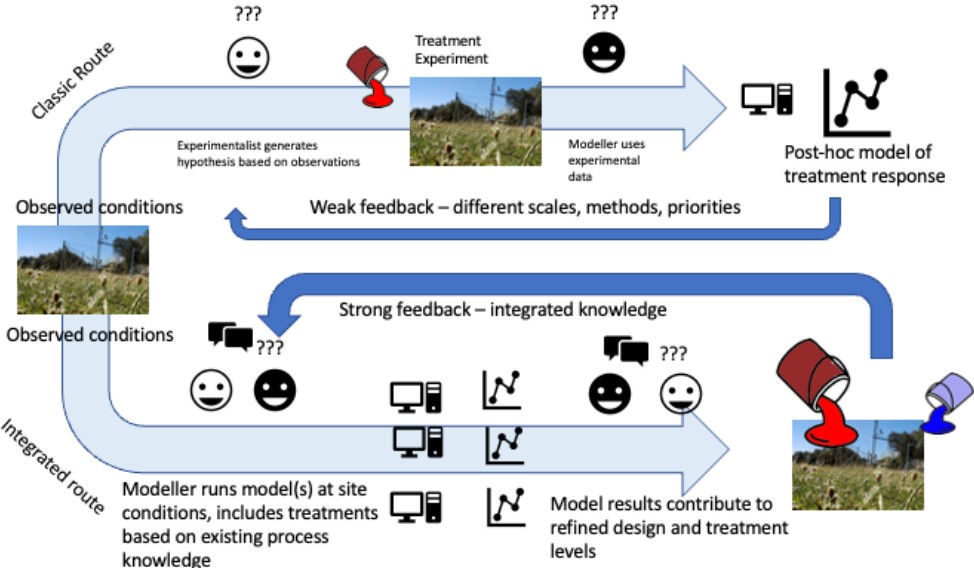

Figure 2 - Classic workflow (A) contrasted with proposed workflow (B) for using models together with ecosystem manipulative
experiments for experimental design and hypothesis generation. Through iterative synthesis of models and experiments in B) and tuning of
experimental parameters based on modelled feedback, stronger predictions are generated.