# Peer review of "Ideas and perspectives: Beyond model evaluation - combining experiments and models to advance terrestrial ecosystem science"

_Biogeosciences, 2023_

## Author Response (AR1)

Dear editor,

Thank you for your work in handling our manuscript. Below you can find the line by line responses to the reviewers comments. Comments are in italics and our replies in regular font. Line numbers refer to the original manuscript. In addition to the changes detailed below, we have made some small cosmetic text changes and updated the paragraph listing contributions to the Special Issue, to include all up to date information on the papers. Additionally, we have changed the citation style to the correct Copernicus formatting following an earlier editorial comment.

Kind regards,

Silvia Caldararu on behalf of the authors.

**Referee #1**

*Overall this is a well written and clear general assessment of opportunities and challenges for experiment-model synergy for increasingly common ecosystem models.*

*This is a topic that is currently of significant interest in the field and the authors do a good job of general summation, if fairly non-specific in places. However I think this is to be expected and a finer look or proposal for a specific approach is perhaps a separate paper or endeavor.*

*My suggestions for improvement are minor and related to small improvements in clarity n the text and figures*

We would like to thank the reviewer for their very positive comments and the time taken to review our manuscript in detail. Below we respond to each comment point by point.

*Specific comments*

*Line 44: Given that the argument here is that data richness, much of which is relatively recent, has not led to decreases in projection uncertainty it may be appropriate to cite more recent refs here (vs 2013/2014)*

We agree that these references are out of date and we have changed them to Arora et a;., 2020 Bastos et al., 2020; Piao et al., 2020.

*Line 73: Not quite sure what 'leverage complementary constraints' means here? Could the authors elaborate?*

We have now clarified this, to read: " Irrespective of what we call them, all such models face similar challenges, in that they need to balance realism, robustness, and reliability (Prentice et al., 2015), as well as computational efficiency to achieve reliable predictions for ecosystem fluxes and pools across the globe and across decades into the future. To achieve this such models need to be confronted with diverse data to leverage complementary constraints. Observations of traits, fluxes, and biomass, obtained at different scales in space and time, and across diverse ecosystem types provide complementary information for calibrating and testing terrestrial biosphere models (Keenan et al., 2012)."

*Line 74: Again, more recent references attesting to model divergence (most recent CMIP maybe?) would support this statement better*

These references have now been changed to Bastos et al., 2020; Friedlingstein et al., 2022,

*Line 149: This feels like two distinct ideas mashed together in the same paragraph, could benefit from a restructure*

We agree that this paragraph was unclear, however it does only contain one idea. It has now been modified to read: "In contrast, models produce all variables consistently at regular intervals, including conceptual or latent quantities which can be expressed mathematically and can be used to quantify and explain ecosystem states, but cannot be measured. These can therefore be used alongside experimental results to explain observed responses. For example, the issue of nutrient limitation under elevated $CO_2$ involves many plant and soil processes, many of which cannot be monitored easily but are included in nutrient-enabled models (Caldararu et al., 2022). One potential issue with this approach is that to have sufficient confidence in these essentially impossible to measure quantities, models need to be able to accurately represent ecosystem processes at a given particular experimental site. Before being used in such a scenario, models should be evaluated or even calibrated for the site using variables that can commonly be measured to make sure of the model performance."

*Line 193: I think there is an unfinished sentence here. "However, trait-based, individual or cohort-based models (Fisher et al., 2015).", so the idea referred to in the following sentence is unclear*

Our apologies. This has now been corrected to: "However, current ecosystem models are not well suited for dealing with changes in species composition, demography, and

competition processes. However, trait-based, individual or cohort-based vegetation demography models (Fisher et al., 2015) have the potential to resolve this limitation."

*Line 234: "Only if an ecosystem's slow biogeochemical cycling is "hit hard" in an experimental setup, underlying processes are revealed." This isn't quite clear. That extensive manipulations unique to EMEs are necessary to differentiate between model formulations?*

Yes, this is the message of the paragraph. It has now been modified to read more clearly: "Only if an ecosystem's slow biogeochemical cycling is "hit hard", with altered conditions in an experimental setup, can underlying processes be revealed. EMEs can thus help to resolve equifinality of model parameter combinations that may not be sufficiently constrained by unperturbed field observations (Keenan et al., 2012) and help avoid cases of getting the 'right answer for the wrong reasons', where different model representations give the same overarching ecosystem level state variables but can lead to different results under altered environmental conditions."

*Line 263: This section isn't too clear. What methods are being referred to re capturing patterns in functional diversity? More details on the methods and maybe an example of a functional trait/characteristic would be helpful*

We have clarified this paragraph as: "One application of note of such proximal sensing data on the horizon is capturing patterns of functional diversity, specifically related to morphological, physiological or phenological traits that can be measured through reflectance. Such data can be validated in the field (Pacheco-Labrador et al., 2022) and through EME explore causative mechanisms behind changes in environment, changes in function and changes in species or functional diversity."

*Line 283: This section is entitled 'new data for EME-Model synthesis' but the paragraph on EC just describes the method and its constraints for data collection, which isn't new per se, without discussing advances in how it could be used for model integration or used as a model for other ecosystem measurements. I don't quite see without those aspects how it fits in this section, until it's used as an example for replication in the next paragraph. I think a different topic sentence here might integrate the EC discussion better.*

This paragraph discusses the potential for EMEs paired with EC sites, which is technically difficult and novel, to the extent that there are only two such experiments currently running. We have now changed the first sentence to make this more clear: "Eddy covariance (EC) data are one of the most frequently used datastreams for model evaluation, parameterization and data assimilation, yet pairings of EMEs with such measurements are rare but could be potentially extremely useful."

*Fig 1. The redundant information in the circle size and color here is a little confusing, and the reader needs to read the legend to appreciate the scale. Perhaps selecting color-blind friendly colors and sticking with one mode (e.g. color only) to convey info with a visual legend would be more effective?*

We have now re-designed Figure 1 for a better colour combination and more intuitive symbols, while maintain the same content

*Figure 2. The legend refers to A and B which do not appear on the figure. More clearly separating the two routes (i.e. by being more spatially separated, making the labels stand out more, or different color schemes for each,, or even separation into separate panels) would improve the interpretability of this figures*

Thank you for your suggestion, we have now clarified the information in the figure

*In addition there are a few typo issues:*

*Line 43: 'and' missing before carbon?*

Corrected

*Line 54: Is there a word missing here 'avoid scaling biases maybe?*

Corrected

*Line 65, no comma needed after traits*

Corrected

*Line 94: I think this was meant to be one sentence, not two.*

Corrected

*Line 96: Is this for a special issue?*

Yes, this is the opening paper for a special issue

*Line 149: Missing a comma*

Sentence changed in response to comment above

**Referee #2**

*This paper describes how experiments and models can be used together in ecological research, and how such an approach can benefit experimental design and interpretation of results. It includes discussions of the different ways that models can be combined with experiments as well as emerging datasets that show promise for improving model-data integration. Overall, the paper was an informative and enjoyable read and presents timely and important ideas for improving model-empirical collaborations. I especially appreciated the discussion of the different ways models can be applied in an experimental context, which I thought was very helpful for getting beyond the often narrow conceptions of what models can be used for. I think this paper is a valuable roadmap for the field.*

Thank you for the very positive assessment of our manuscript.

*I have a few suggestions of areas where the paper's clarity could be improved:*

*Line 45: I think the referencing for this sentence would be more effective if it could include some more recent studies. It's impossible for a paper from 2013 or 2014 to show improvement in models from data products that weren't available until 2017 or 2020 (as cited in the previous few sentences). Are there citations from models that do include some of those advances that can support the statement that uncertainties have not decreased?*

We agree that these references are out of date and we have changed them to Arora et al., 2020, Bastos et al., 2020; Piao et al., 2020.

*Line 70: I might also include computational feasibility as part of the balance – it may be possible to create a highly detailed model that is too computationally expensive to run using available resources.*

Thank you for the suggestion, now added.

*Line 80: The FACE acronym should be spelled out*

Now added.

*Line 113: It would be helpful to explain the "digital twin" concept*

We have now added a definition, as follows: " Models can be used as digital twins (digital representations of physical objects or environments) or sandboxes to explore different possibilities for experimental design, set realistic and practical treatment levels and formulate quantitative, defensible hypotheses."

*Line 135-148: It would be helpful to provide some examples of existing studies that have used this type of approach, if possible.*

To the best of our knowledge, this approach has not yet been documented in a published experiment.

*Line 193: It seems like the second half of this sentence is missing.*

Our apologies. This has now been corrected to: "However, current ecosystem models are not well suited for dealing with changes in species composition, demography, and competition processes. However, trait-based, individual or cohort-based vegetation demography models (Fisher et al., 2015) have the potential to resolve this limitation."

*Line 212: I don't think there is a section 3.3.*

Thank you for spotting this, it has now been corrected to section 2.2.

*Line 286: Another relevant example is the Forest Accelerated Succession Experiment: Gough, C. M., Bohrer, G., Hardiman, B. S., Nave, L. E., Vogel, C. S., Atkins, J. W., Bond-Lamberty, B., Fahey, R. T., Fotis, A. T., Grigri, M. S., Haber, L. T., Ju, Y., Kleinke, C. L., Mathes, K. C., Nadelhoffer, K. J., Stuart-Haëntjens, E., and Curtis, P. S.. 2021. Disturbance-accelerated succession increases the production of a temperate forest. Ecological Applications 31( 7):e02417. 10.1002/eap.2417*

Thank you for this very relevant reference which we had missed. It has now been added to the text.

*Line 295-297: Another relevant example of a large manipulative experiment that used a gradient (or regression) approach to treatments is the SPRUCE experiment: Hanson, P. J., Griffiths, N. A., Iversen, C. M., Norby, R. J., Sebestyen, S. D., Phillips, J. R., et al. (2020). Rapid net carbon loss from a whole-ecosystem warmed Peatland. AGU Advances, 1, e2020AV000163.* [https://doi.org/10.1029/2020AV000163](https://doi.org/10.1029/2020AV000163)

Thank you for the suggestion, we have now added this reference and the paragraph now reads: "Similarly, quantitative gradients allow parameterisation of response surfaces (as opposed to step changes in standard EMEs) but limit possible replication (Kreyling et al., 2018) and has successfully been used in ecosystem scale experiments including a multi-factorial grassland experiment (ClimGrassHydro, Piepho et al., 2017) and a warming peatland experiment (SPRUCE, Hanson et al., 2020)."